# Symptoms after COVID-19 Infection in Individuals with Multiple Sclerosis in Poland

**DOI:** 10.3390/jcm10225225

**Published:** 2021-11-10

**Authors:** Agata Czarnowska, Katarzyna Kapica-Topczewska, Olga Zajkowska, Monika Adamczyk-Sowa, Katarzyna Kubicka-Bączyk, Natalia Niedziela, Paweł Warmus, Alicja Kalinowska-Łyszczarz, Karolina Kania, Agnieszka Słowik, Marcin Wnuk, Monika Marona, Klaudia Nowak, Halina Bartosik-Psujek, Beata Lech, Adam Perenc, Małgorzata Popiel, Marta Kucharska-Lipowska, Monika Chorąży, Joanna Tarasiuk, Anna Mirończuk, Jan Kochanowicz, Anetta Lasek-Bal, Przemysław Puz, Katarzyna Maciejowska, Sławomir Wawrzyniak, Anna Niezgodzińska-Maciejek, Anna Pokryszko-Dragan, Ewa Gruszka, Sławomir Budrewicz, Marta Białek, Jacek Zwiernik, Anna Michałowska, Krzysztof Nosek, Beata Zwiernik, Bożena Lewańczyk, Waldemar Brola, Alina Kułakowska

**Affiliations:** 1Department of Neurology, Medical University of Białystok, M. Skłodowskiej-Curie 24A, 15-276 Białystok, Poland; katarzyna-kapica@wp.pl (K.K.-T.); chorazym@op.pl (M.C.); amirtarasiuk@wp.pl (J.T.); anna-mironczuk@wp.pl (A.M.); kochanowicz@vp.pl (J.K.); alakul@umb.edu.pl (A.K.); 2Faculty of Economic Sciences, University of Warsaw, 00-241 Warszawa, Poland; o.zajkowska@gmail.com; 3Department of Neurology, Faculty of Medical Sciences in Zabrze, Medical University of Silesia, 40-055 Katowice, Poland; m.adamczyk.sowa@gmail.com (M.A.-S.); neurozab@sum.edu.pl (K.K.-B.); natalia@niedziela.org (N.N.); pawel.warmus86@gmail.com (P.W.); 4Department of Neurology, Division of Neurochemistry and Neuropathology, Poznan University of Medical Sciences, 61-701 Poznań, Poland; alicjakal@yahoo.com (A.K.-Ł.); karolina-kania@o2.pl (K.K.); 5Department of Neurology, Jagiellonian University Medical College, University Hospital, 30-688 Krakow, Poland; slowik@neuro.cm-uj.krakow.pl (A.S.); marcin.a.wnuk@gmail.com (M.W.); monika.marona@interia.pl (M.M.); neurologia@cm-uj.krakow.pl (K.N.); 6Department of Neurology, Institute of Medical Sciences, Medical College of Rzeszow University, 35-310 Rzeszów, Poland; bartosikpsujek@op.pl; 7Neurology Clinic with Brain Stroke Sub-Unit, Clinical Hospital No. 2 in Rzeszow, 35-301 Rzeszów, Poland; sekretariat@szpital2.rzeszow.pl (B.L.); lidiaiadam.perenc@wp.pl (A.P.); gosiaur210474@interia.pl (M.P.); 8Department of Neurology, Specialist Hospital in Końskie, 26-200 Końskie, Poland; neurol@zoz.konskie.pl; 9Department of Neurology, School of Health Sciences, Medical University of Silesia, 40-055 Katowice, Poland; alasek@gcm.pl (A.L.-B.); ppuz@o2.pl (P.P.); neurologia@sum.edu.pl (K.M.); 10Department of Neurology, 10th Military Research Hospital and Polyclinic, Independent Public Healthcare Centre, 85-681 Bydgoszcz, Poland; swawrzyniak@wp.pl (S.W.); szpital@10wsk.mil.pl (A.N.-M.); 11Department of Neurology, Wroclaw Medical University, 50-367 Wroclaw, Poland; anna.pokryszko-dragan@umed.wroc.pl (A.P.-D.); ewa.gruszka@umed.wroc.pl (E.G.); slawomir.budrewicz@umed.wroc.pl (S.B.); 12Department of Neurology, Regional Specialised Hospital No. 4 in Bytom, 41-902 Bytom, Poland; marta_ewa@interia.pl; 13Neurology Ward, Provincial Specialist Hospital, 10-561 Olsztyn, Poland; jacekzwiernik@gmail.com (J.Z.); szpital@wss.olsztyn.pl (A.M.); 14Department of Neurology, University of Warmia and Mazury, 10-719 Olsztyn, Poland; beata.zwiernik@uwm.edu.pl; 15Department of Pharmacology and Toxicology, Faculty of Medicine, University of Warmia and Mazury, 10-719 Olsztyn, Poland; krzysztof.nosek@uwm.edu.pl; 16Clinic of Neurology, University of Warmia and Mazury, 10-719 Olsztyn, Poland; 17Neurology Ward, Provincial Integrated Hospital, 82-300 Elbląg, Poland; sekretariat@szpital.elblag.pl; 18Department of Neurology, Specialist Hospital in Końskie, Collegium Medicum, Jan Kochanowski University Kielce, 26-200 Końskie, Poland; wbrola@wp.pl

**Keywords:** multiple sclerosis, COVID-19, SARS-CoV-2, disease modifying therapies, residual symptoms

## Abstract

(1) Background: To report and analyze the presence of residual symptoms after SARS-CoV-2 infection among Polish patients with multiple sclerosis (MS) treated with different disease-modifying therapies (DMTs). (2) Methods: The study included 426 individuals with MS treated with DMTs and confirmed SARS-CoV-2 infection from 12 Polish MS centers. The data were collected through to 31 May 2021. The information included demographics, specific MS characteristics, course of SARS-CoV-2 infection, and residual (general and neurological) symptoms lasting more than four and 12 weeks after the initial infection. The results were obtained using maximum likelihood estimates for odds ratio and logistic regression. (3) Results: A total of 44.84% patients with MS reported symptoms lasting between four and 12 weeks after the initial infection; 24.41% people had symptoms that resolved up to 12 weeks, and 20.42% patients had symptoms that lasted over 12 weeks. The most common symptoms were: fatigue, disturbance of concentration, attention, and memory, cognitive complaints, and headache. None of the DMTs were predisposed to the development of residual symptoms after the initial infection. A total of 11.97% of patients had relapse three months prior or after SARS-CoV-2 infection. (4) Conclusion: Almost half of individuals with MS treated with different DMTs had residual symptoms after SARS-CoV-2 infection. None of the DMTs raised the probability of developing post-acute COVID symptoms.

## 1. Introduction

The pandemic of coronavirus disease 2019 (COVID-19) was declared by World Health Organization in March of 2020 after emerging from Wuhan China a few months earlier [1,2]. Cases have continued to increase. The third global wave overlapped with the introduction of vaccinations against the virus. The optimal effect is not yet achieved yet due to the varying availability of vaccines and immunization strategies for different countries. New variants of the virus are still being discovered some with even more severe clinical outcomes [3].

The severe acute respiratory syndrome coronavirus-2 (SARS-CoV-2) responsible for COVID-19 causes a number of clinical symptoms. Infection is a special thread to those with comorbid diseases and those over 60 years old [4]. Reports of the disease raised concern among neurologists treating autoimmune diseases, including multiple sclerosis (MS). There has been much uncertainty about the course of the disease in individuals undergoing disease modifying therapies (DMTs). The initial studies do not suggest an unfavorable course [5,6]. Several point out a greater risk of severe infection in people treated with anti-CD20 agents, but there are significant limitations (small number of cohorts) to draw any final conclusions. A higher level of disability was associated with worse outcomes [7,8,9].

Most studies concentrate on the clinical course of the disease, which is understandable because the pandemic is still ongoing. During the acute stage of the infection, neurological symptoms are common (in up to 73% of hospitalized patients) [10,11]. Current reports show that many patients struggle with residual deficits lasting weeks or even months after the initial COVID-19 infection [12].

Persistent neurological symptoms after COVID-19 are a special concern for neurologists—especially those who treat patients with immunological disorders like MS. Complications after SARS-CoV-2 infection are not sufficiently well described in individuals with underlying immunological dysregulation or in people with MS receiving DMTs. Determining whether the neurological symptoms are related to an immune disease, treatment, other factors, or are independent complications may be a significant challenge in the future. To better understand the prolonged impact of SARS-CoV-2 infection on individuals with MS, we collected data from 12 Polish MS centers. The goal was to analyze the course of SARS-CoV-2 infection in patients with MS treated with different DMTs and to report the presence of residual symptoms.

## 2. Materials and Methods

The Multiple Sclerosis and Neuroimmunology Section of the Polish Neurological Society published an announcement about the study at www.ptneuro.pl (accessed on 7 November 2021), and every MS center in Poland was invited to participate. Finally, participants were recruited from 12 Polish MS centers. The data was obtained by neurologists using a standardized questionnaire (the same for all MS centers, available in Appendix A).

We included individuals with a confirmed diagnosis of MS according to 2010 and 2017 McDonald criteria: Most were treated with DMTs. Only patients with confirmed SARS-CoV-2 infection were registered for the study. The infection was detected by positive polymerase chain reaction (Seegene Allplex 2019 nCoV assay, PCR, SARS-CoV-2 Real Time PCR LAB-KIT™ by BIOMAXIMA S.A., GeneProof SARS-CoV-2 PCR Kit), positive antigen test against SARS-CoV-2 (Panbio™ COVID-19 Ag Rapid Test, Abbott), or the presence of antibodies against SARS-CoV-2 (EUROIMMUN Anty-SARS-CoV-2 ELISA IgA, IgG). Disability was assessed by the Expanded Disability Status Scale (EDSS) [13]. We collected patient demographics, current DMT use, information about the course of SARS-CoV-2 infection (symptoms, hospitalization, oxygen therapy, pharmacotherapy, and short-term outcome), and presence of residual symptoms after COVID-19. The neurological symptoms after the infection were divided into two categories: (1) symptoms lasting from 4 to 12 weeks after the initial SARS-CoV-2 infection and (2) symptoms lingering over 12 weeks [14]. The data was collected through 31 May 2021. The additional data about COVID-19 cases in the general Polish population were obtained from official reports of the Polish Ministry of Health [15].

Demographics, MS duration, level of disability, DMT use, method of SARS-CoV-2 infection confirmation, symptoms, number of hospitalizations, treatment during SARS-CoV-2 infection, comorbidities, presence of relapse, and all information regarding residual symptoms after the infection (lasting more than 4 and 12 weeks) were reported with descriptive statistics. Metrics included range, mean, and standard deviation or median and interquartile range depending on the data type.

To further evaluate between-group comparisons, we divided the entire cohort into three groups depending on the severity of the primary SARS-CoV-2 infection. Group 1 patients reported 4 and fewer symptoms listed in the survey. Group 2 patients reported more than 4 symptoms listed in the survey, and Group 3 patients required oxygen therapy. Assignment to a higher group implied a more severe disease course. Hospitalization was analyzed separately in created logistic regression models. The division was based on the data obtained. Blood saturation or the result of chest computed tomography (frequently used in other studies) could not be included in the analysis because it was not performed across the entire cohort.

To separately identify risk factors for residual neurological symptoms in the cohort, the study group was divided into individuals who fully recovered, individuals who had symptoms lasting more than 4, and individuals who had symptoms lasting more than 12 weeks. For statistical comparisons, maximum likelihood estimates for odds ratios were calculated to compare risk factors. Further analysis controlled for several potential confounders, and logistic regression models were estimated. To assess the relevance of residual respiratory track symptoms we used Fisher’s exact test, *t* test, Mann–Whitney test and maximum likelihood estimates of the odds ratios. Given the space limit and aiming to focus on most important and significant results we skip most of the estimation output tables or reduce table’s dimensions. All calculations were performed with STATA 15 software (StataCorp 2017) [16].

The study was approved (approval No.6/2021) by the Bioethics Committee at Collegium Medicum, Jan Kochanowski University in Kielce, Poland.

## 3. Results

### 3.1. Demographics and Clinical Characteristic of the Study Group

The study included information about 426 patients with MS and confirmed SARS-CoV-2 infection. Most subjects were treated with different DMTs (99.06%); 280 (65.73%) patients from the cohort were diagnosed with the initial infection in 2020, and 146 (34.27%) cases were recognized in 2021. In our observation (from first reported cases through 31 May 2021), the peak incidence of COVID-19 in people with MS generally overlapped with the general Polish population (Figure 1).

Demographics and clinical characteristic of the study group are presented in Table 1. People in the cohort were treated with following DMTs: interferon beta (77 individuals, 18.08%), glatiramer acetate (43, 10.09%), dimethyl fumarate (171, 40.14%), teriflunomide (34, 7.98%), fingolimod (16, 3.76%), natalizumab (29, 6.81%), ocrelizumab (15, 3.52%), cladribine (seven, 1.64%), alemtuzumab (one, 0.23%), mitoxantrone (one, 0.23%), ozanimod (12, 2.82%), other therapies (16, 3.76%), and few individuals without treatment (four, 0.94%).

The most common comorbid diseases were hypertension (44 people, 10.3%), diabetes (10, 2.3%), and asthma (10, 2.3%).

### 3.2. SARS-CoV-2 Infection

The infection was confirmed by PCR tested on a nasopharyngeal swab in 361 individuals with MS (84.74%). The antigen test was positive in 44 patients (10.33%), and antibodies against SARS-CoV-2 were detected in 24 patients (5.63%). A few patients had a combination of the various tests. Having contact with someone infected was reported by 265 people (62.21%).

The initial infection of SARS-CoV-2 was manifested by following symptoms: fever (222 people, 52.11%), fatigue (179, 42.02%), loss of smell (174, 40.85%), dry cough (156, 36.62%), muscle pain (149, 34.98%), headache (141, 33.1%), bone and joint pain (94, 22.07%), sore throat (84, 19.72%), loss of taste (53, 12.44%), swelling of the nasal mucosa (48, 11.27%), chills (47, 11.03%), dyspnea (32, 7.51%), rash (nine, 2.11%), diarrhea (eight, 1.88%), vomiting (one, 0.23%), and abdominal pain (one, 0.23%); 25 individuals (5.87%) were asymptomatic.

Most patients did not receive any specific treatment during SARS-CoV-2 infection. 71 individuals (16.7%) received antibiotics, 15 (3.5%) glucocorticosteroids, 10 (2.3%) convalescent plasma therapy, and seven (1.6%) were treated with remdesivir. There were 13 (3%) people who required passive oxygen therapy, and only four (0.9%) individuals were treated with mechanical ventilation.

There were 24 people (5.63%) with a relapse at most three months prior to the initial infection, and 27 (6.34%) had a relapse at most three months after the initial infection. Among individuals, who had relapse after the infection, the minimal EDSS increase was 1 point and maximal was 3.5 points (Median 2, Mean 1.85, SD = 0.8528), all were treated with intravenous corticosteroids (metyloprednizolon 3–5 g). Symptoms during the relapse were as following: pyramidal track symptoms (16 people), cerebellar symptoms (eight people), sensory deficit (four people), brainstem symptoms (three people), urinary incontinence (one person). MRI was not performed as standard for all patients; however, it was made in every individual with brainstem or spinal cord symptoms (four individuals: Three had gadolinium enhancing lesions, one had new T2 lesion). The mean time for relapse occurrence after the SARS-CoV-2 infection was 43 days. Patients with relapse up to three months before the initial infection did not predispose for having residual symptoms lasting four to 12 weeks (*p* = 0.530, OR = 1.325, CI = 0.579, 3.031) or over 12 weeks (*p* = 1.000, OR = 1.060, CI = 0.383, 2.935) in comparison to individuals without relapse. Individuals with relapse at most three months after SARS-CoV-2 infection had residual symptoms lasting between four and 12 weeks more often in comparison to people without relapse after the infection, the difference was statistically significant (*p* = 0.041, OR = 2.208, CI = 1.043, 4.673). The correlation was not significant for symptoms lasting over 12 weeks (*p* = 0.261, OR = 1.576, CI = 0.698, 3.556). However, lack of statistical significance in differences might be due to small number of relapses in the sample.

The overall hospitalization rate due to COVID-19 in the cohort was 6.34% (27 patients); 13 people were hospitalized >10 days. Only two individuals (0.47%) died due to COVID-19 infection. The first had meningitis and acute hepatitis failure as SARS-CoV-2 infection complications, the second had respiratory failure.

### 3.3. Residual Symptoms

Most individuals (235 people, 55.16%) did not suffer from residual neurological or general symptoms after the initial infection, 191 (44.84%) people reported symptoms lasting between four and 12 weeks after the initial infection: in 104 (24.41%) individuals, the symptoms resolved within 12 weeks, but 87 (20.42%) had symptoms that lasted over 12 weeks after the initial infection. The distribution of different symptoms is shown in Table 2.

Residual symptoms lasting four to 12 weeks were more frequent in people who presented the following symptoms during initial infection: fever (OR = 1.958, *p* = 0.001, CI = (1.320–2.904)), cough (OR = 1.777, *p* = 0.005, CI = (1.188–2.658)), fatigue (OR = 1.581, *p* = 0.021, CI = (1.069–2.337)), headache (OR = 1.885, *p* = 0.002, CI = (1.247–2.849)), muscle pain (OR = 2.053, *p* = 0.001, CI = (1.362–3.096)), and loss of smell (OR = 2.117, *p* = 0.000, CI = (1.419–3.160)). Among individuals with symptoms lingering over 12 weeks, we observed more frequently the following prior symptoms: fatigue (OR = 1.995, *p* = 0.019, CI = 1.108–3.592), and loss of taste (OR = 2.626, *p* = 0.019, CI = (1.132–6.091)).

We estimated logistic regressions for the probability of developing residual symptoms lasting four to 12 weeks and separate logistic regression for developing symptoms lingering over 12 weeks (conditional on having symptoms lasting >four weeks). The following confounders were considered: age, gender, body mass index (BMI), EDSS, duration of the disease, smoking, and lymphopenia. Only age was predisposed patients to developing symptoms lasting four to 12 weeks (OR varied from 1.041 to 1.044, where *p* varied from 0.002 to 0.007). Several different specifications with different subsets of confounding variables were estimated. Patients’ hospitalization, if included into the model, raised the chances for having residual neurological symptoms (OR 4.542, *p* = 0.004, CI = 1.601–12.890); the same tendency was observed in augmented model for people who needed oxygen therapy (OR = 9.483, *p* = 0.004, CI = 2.010–44.734).

Separate logistic regressions were estimated to assess the likelihood of hospitalization, oxygen therapy, and residual fatigue. The included confounders were gender, age, BMI, EDSS, duration of the disease, smoking, and lymphopenia. According to our estimates each subsequent year of age increases the odds of developing fatigue by 3.7%. Furthermore, the need for oxygen therapy increases the chances of developing residual symptoms by 12 times. Finally, increasing EDSS by one point raises the chance of hospitalization by 69.8%.

Among all patients with residual symptoms, 26.7% had those related to respiratory track. This group of patients had a higher need of hospitalization during the initial infection (*p*= 0.018, OR = 3.225, CI = 1.276, 8.151). The development of respiratory track residual symptoms were not dependent on sex (*p* = 0.603), age (*p* = 0.982) or patients EDSS (*p* = 0.982). The average age of individuals with pulmonary symptoms were on average 0.821 year older. Having relapse prior or after COVID-19 did not raised chances for residual respiratory track symptoms (*p*= 0.308, OR = 2.073, CI = 0.618, 6.952 and *p*= 1.000, OR = 0.967, CI = 0.330, 2.836). There was no significant statistical correlation between residual respiratory track symptoms and any particular neurological residual symptoms.

Among comorbid diseases, only asthma was predisposed for post COVID symptoms in the study group (OR = 5.093, *p* = 0.024, CI = 1.057–24.550).

### 3.4. Specific Residual Neurological Symptoms and MS

We found that higher EDSS was a predisposing factor for having a consequent residual headache (OR = 1.372, *p* = 0.026, CI = 1.039–1.813); age was a predisposing factor for having dizziness after infection (OR = 1.073, *p* = 0.017, CI = 1.013–1.137), and longer disease duration was a predisposing factor for having residual olfactory deficit (OR = 1.073 for each additional year, *p* = 0.049, CI = 1.000–1.151). The results were obtained based on logistic regression. Separate regressions were estimated for each symptom. We controlled for several confounders including age, gender, BMI, smoking, lymphopenia, and COVID severity. For robustness check, we estimated our models using several subsets of potential confounding variables. Selected estimates are shown in Table 3.

### 3.5. DMTs and Residual Symptoms after SARS-CoV-2 Infection

Maximum likelihood odds ratios were estimated to assess the impact of different DMTs. Neither therapy is predisposed to the development of residual symptoms after the initial infection. However, individuals treated with fingolimod had a higher chance of developing symptoms lasting over 12 weeks conditional on having symptoms lasting four to 12 weeks (OR = 4.462, *p* = 0.047, CI = 0.884–22.539).

## 4. Discussion

This study is the largest analysis of Polish people with MS treated with DMTs after SARS-CoV-2 infection, with a focus on the presence of residual neurological symptoms. The mechanism behind the residual neurological complications after SARS-CoV-2 infection is still under investigation. There are ongoing considerations if the reported symptoms are induced by the virus itself or trigger a chain of immune reactions. Studies show that the nervous system presumably can be affected in multiple ways both directly and indirectly [17,18].

Numerous recent publications (including our recent publication concerning patients with MS) conclude that the course of SARS-CoV-2 infection in individuals with MS is generally favorable [6,19,20]. There is no proof that most DMTs have a negative impact on the course of the diseases. However, being under anti-CD20 therapies was shown to be a risk factor for hospitalization, with higher mortality in some studies; this was contradicted in others [7,21,22]. Lower humoral response after COVID-19 was observed for people treated with anti-CD20 agents and fingolimod in the work of Bigaut et al. [23].We know that vaccination has to be tailored to different DMTs [24].

Our data consist almost exclusively of individuals presenting at least one symptom of infection. Isolated asymptomatic cases were detected for professional/travel reasons or before planned hospitalizations. Almost half of the cohort presented residual symptoms after initial infection (44.84%), but most people recovered fully in less than 12 weeks. Symptoms considered as residual after COVID-19 were as follows: fever, pain, fatigue, cough, dyspnea, chest pain, palpitations, gastrointestinal symptoms muscle pain, bones/joint pain, cognitive complaints, disturbance of concentration, attention, and memory, headache, sleeping difficulties, peripheral neuropathy, vertigo, disturbance of smell/taste, depression, anxiety. The following symptoms could not be present prior to SARS-CoV-2 infection to be included. If a relapse has occurred during the recovery period after COVID-19, it was defined as occurrence of new or worsening of old symptoms classically present as pyramidal, cerebellar, sensory, brain steam or spinal cord deficit, lasting >24 h, always assessed by an experienced neurologist.

Fatigue was the leading issue in most individuals. This observation is consistent with other studies focused on long and short term observation of patients who recovered after COVID-19 infection beyond patients with MS. Fatigue was a leading symptom in observations made even up to 6 months: 46.6% of 384 patients in an average 60-day observation by Mandal et al., 87% of 2113 patients in a 79-day observation by Goertz et al., and 63% of 1733 patients in a six-month observation by Hunag et al. [25,26,27]. Fatigue is also a symptom present in many people with MS regardless of COVID-19 infection [28]. It would be very interesting to evaluate in further studies if this symptom persists after SARs-CoV-2 infection in the same percentage of individuals with MS versus the general population in much longer observations. In short observation of exclusively hospitalized patients, during post-acute faze of the infection (eight to 12 weeks after hospital admission), Arnold et al. also reported fatigue as one of the leading issues (39%). Breathlessness and insomnia were more common in comparison to our cohort [29]. Dyspnea was also more common (34.4%) in the study of Moreno-Pérez et al. in an eight to 12 weeks observation in relation to four to 12 weeks observation of our cohort [30]. The second most common residual symptoms were disturbance of concentration, attention, and memory; cognitive complaints; and headache. The number of patients with these deficits varies dramatically, but these disorders are difficult to define. In most publications, regarding residual symptoms after COVID-19 infection, those are the leading ones [10,31]. The link between SARS-CoV-2 infection and cognitive changes is now widely discussed [32,33]. We found few correlations on particular features of MS: Higher EDSS was a predisposing factor for having a consequent residual headache. Longer disease duration was a predisposing factor for having a residual olfactory deficit. There is no other similar work published to date giving us the opportunity to compare our findings. It would be very interesting to see if these correlations exist in the greater cohort. Dyspnea, anxiety, and depression as residual symptoms were less common in our cohort versus the general population in literature [34,35].

The main focus of our work was to assess if specific MS features and particular DMTs were an important factor in developing residual symptoms after initial SARS-CoV-2 infection. In univariate analysis, we used models focused on characteristic features of MS (EDSS, duration of the disease, relapse occurrence, lymphopenia) and general features (age, gender, BMI, smoking). Here, only age was a relevant risk factor increasing the probability of having residual symptoms. In our previous work, age was identified as an important factor of severe course of the initial infection [36]. Beyond prolonged symptoms, every point of EDSS significantly increased the chance for hospitalization in our cohort. A similar observation was made by other MS centers [37]. The statistical difference in the occurrence of residual symptoms between RRMS and progressive phenotypes of the disease was not significant. The number of people with progressive MS was limited in our cohort, and further work is critical to making any conclusions.

The key observation of our work was made by adding DMTs to designed models. None of the analyzed drugs significantly increased the chance of developing post-infection symptoms. Anti-CD20 agents were also not relevant in case of developing residual symptoms, but they did raise the risk of hospitalization in our previous work and few others published studies [7]. We found that individuals treated with fingolimod were more prone to develop symptoms lasting over 12 weeks (regardless of lymphopenia), but only under the condition of having symptoms lasting four to 12 weeks. One limitation is the small number of individuals treated with fingolimod and the uneven distribution of DMTs in the cohort. However, our observation draws attention to certain correlations for future research. Among comorbid diseases, only asthma predisposed patients to residual symptoms. However, our study group was relatively young with few comorbidities.

A total of 6.34% of individuals with MS had relapse at most three months after SARS-CoV-2 infection in the presented cohort. There are evidence emphasizing the role of certain peripheral infection, especially viral, in triggering relapse in people with MS [38]. There are no long-term observations on correlation between COVID-19 infection and relapse rate. The initial findings are not consistent among studies. Etemadifar et al. did not show an evident increase of relapse incidence in individuals with MS after SARS-CoV-2 infection [39]. However, Barzegar et al. based on their research conclude that COVID-19 infection can be a trigger exacerbation of MS [40]. Due to a small relapse rate in our cohort, we do not draw any conclusions based on our findings.

A potential source of bias in our study is that we did not include immobile individuals with a high degree of motor and/or cognitive disability. The chosen methodology of recruiting the patients limited the cohort to selected MS centers willing to participate, so a significant amount of people with MS from Poland was not reported. The data was collected by multiple neurologist and there is a possibility there were minor differences in the interpretation of the reported symptoms. We could not appreciate the differences between treated versus non-treated people with MS, because our study group almost exclusively included individuals on DMTs (with uneven distribution). Other possible limitation are the retrospective features of the study, in which most of the data concerning symptoms have come from collecting medical history during planned visits or telephone interviews with the patients. These were often done a few weeks after COVID-19; thus, some people might have difficulty remembering the exact course of infection and treatment. What is more, a major limitation of our study is lack of the control group. The significance of relapses after COVID-19 infection could not be appreciated as we did not collected data regarding the previous relapse rate in individuals with MS from the cohort. We are also considering the possibility that relapses after COVID-19 infection could have been misdiagnosed as residual symptoms and vice versa in some individuals, despite a careful neurological assessment.

## 5. Conclusions

Almost half of the individuals with MS treated with different DMTs had residual symptoms after SARS-CoV-2 infection. Fatigue was the most common one. None of the DMTs raised the probability of developing residual symptoms after the initial infection. Age independently increased the probability of having any residual symptoms. The need for oxygen therapy during acute SARS-CoV-2 infection significantly increased the chances of developing prolonged symptoms. Higher EDSS was a predisposing factor for having consequent residual headache, and longer disease duration was a predisposing factor for having a residual olfactory deficit. It is difficult to draw final conclusions because of the limited number of patients, but our results are an interesting starting point for further research.

## Figures and Tables

**Figure 1 jcm-10-05225-f001:**
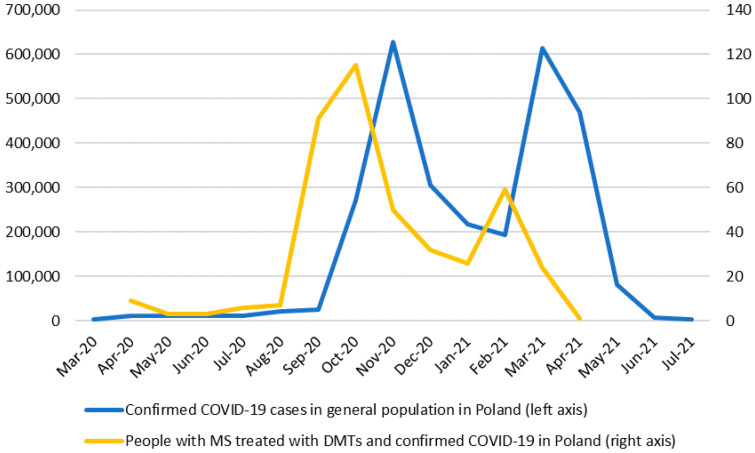
Confirmed cases of COVID-19: Individuals with MS versus general Polish population. Source: Ministry of Health, our own database of patients with MS, and our own calculations.

**Table 1 jcm-10-05225-t001:** Demographics and clinical characteristic of patients with MS.

Demographics	No.	(%)	Range	Mean	MEDIAN	IQR	SD
Age			20–68	40.27	40	15	10.12
Male	142	33.3					
Female	284	66.7					
Smokers	40	9.39					
	Clinical characteristic due to MS						
Disease course							
RRMS PPMS SPMS	397 17 12	93.19 3.99 2.82					
EDSS			0–7	2.61	2.5	2	1.40
Disease duration (years)			0–32	8.42	7	8	5.75
Duration of DMTs use (years)			0–16	4.79	4	5	3.61

Abbreviations: MS, multiple sclerosis; SD, standard deviation; IQR, interquartile range; RRMS, relapsing remitting multiple sclerosis; PPMS, primary progressive multiple sclerosis; SPMS, secondary progressive multiple sclerosis; EDSS, the expanded disability status scale; DMTs, disease-modifying therapies.

**Table 2 jcm-10-05225-t002:** Frequency of residual symptoms in the group of individuals with MS after SARS-CoV-2 infection.

	Symptoms Lasting 4–12 Weeks N (% of the Group with Symptoms Lasting 4–12 Weeks)	Symptoms Lingering >12 Weeks N (% of the Group with Symptoms >12 Weeks)	Percentage of the Whole Sample
General symptoms:			
Fever	0	0	0
Pain	9 (4.71)	5 (5.75)	2.11
Fatigue	107 (56.02)	58 (66.67)	25.12
Cough	37 (19.37)	11 (12.64)	8.69
Dyspnea	24 (12.57)	16 (18.39)	5.63
Chest pain	12 (6.28)	7 (8.05)	2.81
Palpitations	22 (11.52)	13 (14.94)	5.16
Gastrointestinal symptoms	12 (6.28)	8 (9.2)	2.81
Musculoskeletal symptoms: Muscle pain, bones/joint pain	44 (23.04) 33 (17.28)	22 (25.29) 19 (21.84)	10.33 7.75
Neurological symptoms:			
Cognitive complaints	48 (25.13)	32 (36.78)	11.27
Disturbance of: concentration, attention, memory	72 (37.7)	35 (40.23)	16.9
Headache	51 (26.7)	20 (22.99)	11.97
Sleeping difficulties	35 (18.32)	20 (22.99)	8.22
Peripheral neuropathy	13 (3.05)	10 (11.49)	3.05
Vertigo	23 (12.04)	15 (17.24)	5.4
Disturbances of smell or/and taste	32 (16.75)	18 (20.69)	7.51
Depression	8 (4.19)	8 (9.2)	1.88
Anxiety	15 (7.85)	11 (12.64)	3.52

**Table 3 jcm-10-05225-t003:** Risk factors associated with different neurological symptoms—selected coefficients of logistic regression estimations.

Selected Variables	Symptoms (Dependent Variables)								
	Cognitive Complaints	Disturbance of: Concentration, Attention, Memory	Headache	Sleeping Difficulties	Peripheral Neuropathy	Vertigo	Disturbances of Smell or/and Taste	Depression	Anxiety
**Age OR**	1.022	1.037 *	1.013	1.045	1.037	1.073 *	1.01	1.05	1.067
**Age *p*-value**	(0.284)	(0.043)	(0.527)	(0.068)	(0.451)	(0.017)	(0.673)	(0.248)	(0.065)
**Gender OR**	2163	1.29	1.444	0.642	1.146	0.841	0.908	2.125	0.927
**Gender** ***p*-value**	(0.080)	(0.453)	(0.354)	(0.335)	(0.889)	(0.767)	(0.815)	(0.508)	(0.913)
**EDSS OR**	0.912	1.048	1.372 *	1.263	0.475	1.033	0.739	0.946	0.866
**EDSS *p*-value**	(0.527)	(0.704)	(0.026)	(0.179)	(0.054)	(0.886)	(0.069)	(0.874)	(0.588)
**Duration of the disease OR**	0.943	0.986	0.951	0.995	1.061	0.901	1.073 *	1.044	0.983
**Duration of the disease** ***p*-value**	(0.129)	(0.616)	(0.163)	(0.907)	(0.469)	(0.083)	(0.049)	(0.541)	(0.773)
**Lymphopenia OR**	0.777	1.055	0.931	1.321	1.086	0.516	0.842	1	1
**Lymphopenia *p*-value**	(0.617)	(0.896)	(0.880)	(0.610)	(0.943)	(0.422)	(0.763)	(.)	(.)
**Disease severity-group 2** **OR**	2.883 **	5.061 ***	2.785 **	2.958 *	2.027	0.973	1.763	2.627	3.273
**Disease severity-group 2** ***p*-value**	(0.006)	(0.000)	(0.008)	(0.025)	(0.462)	(0.969)	(0.189)	(0.248)	(0.079)
**Disease severity-group 3** **OR**	14.630 ***	5.332 **	3.815 *	8.822 **	33.670 **	11.60 **	2.035	1.000	14.990 **
**Disease severity-group 3** ***p*-value**	(0.000)	(0.005)	(0.029)	(0.002)	(0.004)	(0.001)	(0.410)	(.)	(0.004)

Abbreviations: OR, odds ratio; EDSS, the expanded disability status scale, * *p* < 0.05, ** *p* < 0.01, *** *p* < 0.001.

## Data Availability

The analysis was bundled into an open R package accessible at https://github.com/Aczarnowska/COVID (accessed on 7 November 2021).

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
