# Peer review of "Symptoms after COVID-19 Infection in Individuals with Multiple Sclerosis in Poland"

_jcm, 2021, doi:10.3390/jcm10225225_

Round 1

Reviewer 1 Report

The paper is interesting, having collected several data. It can provide important results.

However, the statistical analysis is unclear, confused the results presented in the table. 

Author Response

Response to Reviewer 1 Comments

Point 1:

The paper is interesting, having collected several data. It can provide important results.

However, the statistical analysis is unclear, confused the results presented in the table. 

Response 1: Thank you very much for your valuable comment. To provide more clear presentation of the results we simplified the presentation of the results.

We added an information in Material and methods , as follows:

‘Given the space limit and aiming to focus on most important and significant results we skip most of the estimation output tables or reduce table’s dimensions.’

Table 1 was simplified. Information about DMTs was moved into the text.

For more clarity we keep the convention of providing percents up to 2 decimal places and estimates up to 3 decimal places. As follows:

‘Long COVID symptoms were more frequent in patients who presented the fol-lowing symptoms during initial infection: fever (OR=1.958, p=0.001, CI=(1.320-2.904)), cough (OR=1.777, p=0.005, CI=(1.188-2.658)), fatigue (OR=1.581, p=0.021, CI=(1.069-2.337)), headache (OR=1.885, p=0.002, CI=(1.247-2.849)), muscle pain (OR=2.053, p=0.001, CI=(1.362-3.096)), and loss of smell (OR=2.117, p=0.000, CI=(1.419-3.160)). Among patients with post-COVID symptoms we observed more frequently the following prior symptoms: fatigue (OR=1.995, p=0.019, CI=1.108-3.592) and loss of taste (OR=2.626, p=0.019, CI=(1.132-6.091)).

The most common comorbid diseases were hypertension (44 patients, 10.3%), di-abetes (10 patients, 2.3%), and asthma (10 patients, 2.3%). Only asthma was predis-posed for long COVID in the study group among all comorbid diseases (OR=5.093, p=0.024, CI=(1.057-24.550)). 

We estimated logistic regression for the probability of developing long COVID and separate logistic regression for developing post-COVID symptoms (conditional on having long COVID). The following confounders were considered: age, gender, body mass index (BMI), EDSS, duration of the disease, smoking, and lymphopenia. Only age was predisposed patients to developing long COVID symptoms (OR varied from 1.041 to 1.044, where p varied from 0.002 to 0.007). Several different specifications were es-timated. Further addition of hospitalized patients to the model raised the chances for residual neurological symptoms (OR 4.542, p=0.004, CI=(1.601-12.890)); the same ten-dency was observed in expanded model for patients who needed oxygen therapy (OR=9.483, p=0.004, CI=(2.010-44.734)).’

Table 3. : Since we present only selected variables we remove some of the variables to increase clarity of our reasoning.

Reviewer 2 Report

Czarnowska et al. reported residual symptoms after SARS-CoV-2 infection among MS Polish patients. The sample size is appropriate (more than 400 cases), and half of them had  symptoms lasting between 4 and 12 weeks. 

I think the paper is original and could be relevant for MS clinical practice. 

Although, some points could be discussed: 

  • Despite reported by Raveendran et al (doi:10.1016/j.dsx.2021.04.007), is there another reported definition for “ symptoms lasting between 4 and 12 weeks after the initial infection” instead of “ LONG Covid”? It could be misinterpreted as long positivity to the virus, which is not the point of the paper.
  • Similarly could you change with other definitions for “brain fog”?
  • The questionnaire used to assess symptoms (line 104) should be added as supplementary material
  • “we divided the entire cohort into three groups depending on the severity of the primary SARS-CoV-2 infection”. Please add to the text the most common symptoms among groups. 
  • Table1 could be simplify, ie DMTs list could be move into the text, line 1 “n of patients” could be removed
  • Table 3 is difficult to be read, i.e. maybe OR and p could stay in the same line. 
  • "The most common comorbid diseases" should be moved elsewhere in the manuscript
  • Paragraph from line 210 to 213 must be clarify, paragraphs from 3.3 to 3.5 could be sum up as quite redundant. 
  • The first part of the discussion is a bit unfocused on results.
  • line 168 and following: too many repetitions of “patients”

Author Response

Response to Reviewer 2 Comments

Point 1: Despite reported by Raveendran et al (doi:10.1016/j.dsx.2021.04.007), is there another reported definition for “ symptoms lasting between 4 and 12 weeks after the initial infection” instead of “ LONG Covid”? It could be misinterpreted as long positivity to the virus, which is not the point of the paper.

Response 1: Thank you very much for your valuable comment. We know that the definitions in the literature vary among authors. To provide as much clarity as we could the definition used in our work is described in material and methods. The timeline – up to 4 weeks for acute infection and more than 4 and 12 weeks is what we found most commonly in the literature and also in NICE guidelines.  

The term ‘long COVID’ was also used to simplify the manuscript text.

Point 2: Similarly could you change with other definitions for “brain fog”?

Response 2: Thank you very much for your valuable comment. For more clarity we added: ‘cognitive complaints’

Point 3: The questionnaire used to assess symptoms (line 104) should be added as supplementary material

Response3 : Thank you very much for your valuable comment. We will ask the publisher to include the questionnaire used to assess symptoms.

Point 4: “we divided the entire cohort into three groups depending on the severity of the primary SARS-CoV-2 infection”. Please add to the text the most common symptoms among groups.

Response 4: Thank you very much for your valuable comment. Unfortunately, the most common symptoms were quite the same in all groups. What distinguished individual groups was poly-symptomism and additional occurrence of less frequent symptoms.

Point 5: Table1 could be simplify, ie DMTs list could be move into the text, line 1 “n of patients” could be removed

Response 5: Thank you very much for your valuable comment. We changed it as suggested.

Point 6: Table 3 is difficult to be read, i.e. maybe OR and p could stay in the same line.

Response 6: Thank you very much for your valuable comment. Since we present only selected variables we remove some of the variables to increase clarity of our reasoning.

Point 7: "The most common comorbid diseases" should be moved elsewhere in the manuscript

Response 7: Thank you very much for your valuable comment. We replaced it as suggested.

Point 8: Paragraph from line 210 to 213 must be clarify, paragraphs from 3.3 to 3.5 could be sum up as quite redundant.

Response 8: Thank you very much for your valuable comment. Paragraphs 3.3 and 3.5 were sum up together. The paragraph regarding DMTs was left separately. The paragraph containing the mentioned lines (210-213 in the original document) was rewritten to provide more clarity and highlight the fact of using different models.

Point 8: The first part of the discussion is a bit unfocused on results. line 168 and following: too many repetitions of “patients”

Response 8: Thank you very much for your valuable comment. We were hoping to provide a gentle introduction for our discussion section. The repetition of the word ‘patient’ was reduced in mentioned part of the manuscript.

Reviewer 3 Report

a well written report on findings of Sars-Covid-19 infection in a large sample of MS patients from data bases in Poland demonstrating solid results and diminishing fears that DMTs could worsen disease course

Thank you for initing me to comment on this interesting study presented to your Journal of Clinical Medicine
This article "Long COVID-19 and post COVID-19 symptoms in patients with multiple sclerosis treated with disease modifying therapies in Poland" by Agata Czarnowska et al
is an interesting study of a large number ppf MS patients with Covid-19 infection, carefully observed and examined.
It is an important topic because too many MS patients feel uncertain about the infection itself or the vaccination to prevent it. This study provieds useful and relevant data to reduce anxities in this respect
It is a  relevant and interesting question, diligently answered and the topic is indeed original. There is too little research done in this aerea as yet and therefore this work is relevant and indeed needed
The study is performed according to standards and the paper is well written,  the text is clear and easy to read. Conclusions are consistent with the evidence and arguments presented. They do address the main question posed

Author Response

Response to Reviewer 3 Comments

Point 1: a well written report on findings of Sars-Covid-19 infection in a large sample of MS patients from data bases in Poland demonstrating solid results and diminishing fears that DMTs could worsen disease course

Response 1: Thank you very much for your valuable comment.

Reviewer 4 Report

This study provides valuable insights to the COVID19 course in patients with MS. The number of patients and the follow-up are a major asset, whereas the lack of a control group must be seen as a drawback. A couple of compulsory issues, however, need to be clarified or corrected:

  • 27 patients of this cohort had a relapse within 3 months from COVID-19. There is increasing evidence that SARS-CoV-2 infection can trigger relapses or progression. Please reanalyse the data and provide insights to this emerging issue: e.g. when did the release occur, increase of EDSS, relapse symptoms, Gd enhancement on MRI, use of steroids.
  • A clearer distinction of COViD-19 related residual symptoms and relapse related symptoms needs to be made in the discussion.
  • Details on the circumstances of the two patients who died from COVID-19 needs to be added.
  • There is a clear methodological bias for patient selection and reporting. Acknowledge this in the limitation section. Also add further limitations.
  • Rephrase the abstract following revision of the manuscript according to the aforementioned issues.

Author Response

Response to Reviewer 4 Comments

Point 1: 27 patients of this cohort had a relapse within 3 months from COVID-19. There is increasing evidence that SARS-CoV-2 infection can trigger relapses or progression. Please reanalyse the data and provide insights to this emerging issue: e.g. when did the release occur, increase of EDSS, relapse symptoms, Gd enhancement on MRI, use of steroids.

Response 1: Thank you very much for your valuable comment. The information about the relapses are very important. We provided as much additional information as we could. The following text was added to the manuscript:

‘Among individuals, who had relapse after the infection, the minimal EDSS increase was 1 point and maximal was 3.5 points (Median 2, Mean 1.85, SD=0.8528), all patients were treated with intravenous corticosteroids (metyloprednizolon 3-5g). Symptoms during the relapse were as following: pyramidal track symptoms (16 patients), cerebellar symptoms (8 patients), sensory deficit (4 patients), brainstem symptoms (3 patients), urinary incontinence (1 patient). MRI was not performed as standard for all patients, however it was made in every patient with brainstem or spinal cord symptoms (4 individuals: 3 had gadolinium enhancing lesions, 1 had new T2 lesion). The mean time for relapse occurrence after the SARS-CoV-2 infection was 43 days.’

Point 2: A clearer distinction of COViD-19 related residual symptoms and relapse related symptoms needs to be made in the discussion.

Response 2: Thank you very much for your valuable comment. To provide more clarification we added following text to the discussion section:

‘Symptoms considered as post/long COVID were as follows: fever, pain, fatigue, cough, dyspnoea, chest pain, palpitations, gastrointestinal symptoms muscle pain, bones/joint pain, ‘brain fog’, disturbance of concentration, attention, and memory, headache, sleeping difficulties, peripheral neuropathy, vertigo, disturbance of smell/taste, de-pression, anxiety. The following symptoms could not be present prior to SARS-CoV-2 infection to be included. If a relapse has occurred during the recovery period after COVID-19, it was defined as occurrence of new or worsening of old symptoms classi-cally present as pyramidal, cerebellar, sensory, brain steam or spinal cord deficit, lasting >24 hours, always assessed by an experienced neurologist.’

Point 3: Details on the circumstances of the two patients who died from COVID-19 needs to be added.

Thank you very much for your valuable comment. We added information about the cause of death of both patients:

‘The first patient had meningitis and acute hepatitis failure as SARS-CoV-2 infection complication, the second patient had respiratory failure.’

The clinical history behind the first patient is quite complicated, however the direct cause of death was acute liver failure.

Point 4: There is a clear methodological bias for patient selection and reporting. Acknowledge this in the limitation section. Also add further limitations

Thank you very much for your valuable comment. We modified the limitation section, we agree that this matter is very important to acknowledge.

‘A potential source of bias in our study is that we did not include immobile patients with a high degree of motor and/or cognitive disability. The chosen methodology of recruiting the patients limited the cohort to selected MS Centres willing to participate, so significant amount of patients with MS from Poland was not reported. The data was collected by multiple neurologist and there is a possibility there were minor differences in the interpretation of the reported symptoms. We could not appreciate the differences between treated versus non-treated MS patients because our study group almost exclusively included patients on DMTs (with uneven distribution). Other possible limitation is the retrospective features of the study in which most of the data concerning symptoms have come from collecting medical history during planned visits or telephone interviews with the patients. These were often done a few weeks after COVID-19; thus, some patients might have difficulty remembering the exact course of infection and treatment. What is more a major limitation of our study is lack of the control group.’ 

Point 5: Rephrase the abstract following revision of the manuscript according to the aforementioned issues.

Thank you very much for your valuable comment. We verified the abstract's compliance with the content of the revised manuscript.

Round 2

Reviewer 1 Report

The paper has been reviewed, but applied statistical analysis and methodology must be completely revised since it is not clear:  tables are confused, missing confidence intervals for OR, dependent variable not identified. It is not clear if a separate logistic regression model was run for each symptom.

I am sure the authors have at disposal a lot of interesting data, they have prepared a good introduction and probably they can discuss their results very well. However, analysis and exposure of results must be absolutely improved.

Author Response

Response to Reviewer 1 Comments (2)

Point 1:The paper has been reviewed, but applied statistical analysis and methodology must be completely revised since it is not clear:  tables are confused, missing confidence intervals for OR, dependent variable not identified. It is not clear if a separate logistic regression model was run for each symptom.

I am sure the authors have at disposal a lot of interesting data, they have prepared a good introduction and probably they can discuss their results very well. However, analysis and exposure of results must be absolutely improved

Response 1: Thank you very much for your valuable comment.

Tables are confused.

Response: We rearranged table 2 and 3 for more clarity.

Missing confidence intervals for OR.

Response: Missing confidence intervals were supplement.

Dependent variable not identified.

Response: We marked them in table 3 and completed the information in the text.

It is not clear if a separate logistic regression model was run for each symptom.

Response: Separate logistic regression model was run for each symptom, we clarified this in the text. We did our best to provide synthetic description of the results of several different models we estimated in the process of data investigation.

Reviewer 4 Report

Some of the issues raised during the first round were solved. Attention is still required for:

  • title: needs to be rephrased for clarify qne redundancy
  • abstract: the main findings are not clearly presented, the relapse issue is also missing.
  • relapses following COVID-19. Discuss own findings in the context of the literature. Also, was this group removed from the long COVID cohort?
  • long COVID. Discuss own findings in the cotext of the literature
  • use the terms people or individuals with MS instead of patients with MS/MS patients

Author Response

Response to Reviewer 4 Comments

Point 1: title: needs to be rephrased for clarify qne redundancy

Response 1: Thank you very much for your valuable comment. The title is now changed to :

‘Symptoms after COVID-19 infection in individuals with multiple sclerosis in Poland.’

Point 2:abstract: the main findings are not clearly presented, the relapse issue is also missing.

Response 2: Thank you very much for your valuable comment. The numbers showing results in the abstract were removed (only percentage were left) for more clarity and to make the abstract easy to read. The information about relapse was added.

Point 3:relapses following COVID-19. Discuss own findings in the context of the literature. Also, was this group removed from the long COVID cohort?

Response 3: Thank you very much for your valuable comment. We added information about relapses in the discussion and added literature:

‘6.34% of individuals with MS had relapse at most 3 months after SARS-CoV-2 in-fection in presented cohort. There are evidence emphasizing the role of certain periph-eral infection, especially viral, in triggering relapse in people with MS.[41] There are no long-term observations on correlation between COVID-19 infection and relapse rate. The initial findings are not consistent among studies. Etemadifar et al. did not show an evident increase of relapse incidence in individuals with MS after SARS-CoV-2 infec-tion.[42] However, Barzegar et al. based on their research conclude that COVID-19 in-fection can be a trigger exacerbation of MS.[43]’

Unfortunately, we could not compare our findings with a control group (one of the limitation of our study) and we did not collected data regarding the relapse rate in individuals with MS before COVID-19 infection. The following information was added to the limitation section:

‘ The significance of relapses after COVID-19 infection could not be appreciated as we did not collected data regarding previous relapse rate in individuals with MS from the cohort.’

Individuals with relapses were included in the analysis of post COVID symptoms. It is mentioned in the manuscript as follows: ‘majority developed long/post COVID symptoms, but it was not statistically significant’. The evaluation of relapse and symptoms after COVID infection were evaluated by neurologist in every case. Symptoms like fatigue few days after relapse were not considered as residual symptoms after COVID. However fatigue lasting weeks after COVID infection (prior to relapse and after if one occurred were considered as COVID residual symptoms). All clinical scenarios were assessed by neurologist who made the report on patient. 

Point 4:long COVID. Discuss own findings in the cotext of the literature

Thank you very much for your valuable comment. The following section of discussion was expanded:

‘Fatigue was the leading issue in most patientsindividuals. This observation is consistent with other studies focused on long and short -term observation of patients who recov-ered after COVID-19 infection beyond patients people with MS. Fatigue was a leading symptom in observations made even up to 6 months: 46.6% of 384 patients in an average 60-day observation by Mandal et al., 87% of 2113 patients in a 79-day observation by Goertz et al., and 63% of 1733 patients in a 6-month observation by Hunag et al.[30][31][32][30][31][32]. Fatigue is also a symptom present in many patients people with MS regardless of COVID-19 infection[33][33]. It would be very interesting to evaluate in further studies if this symptom persists after SARs-CoV-2 infection in the same percentage of patients individuals with MS versus the general population in much longer observations. In short observation of exclusively hospitalized patients, during post-acute faze of the infection (8–12 weeks after hospital admission), Arnold et al. also reported fatigue as one of the leading issues (39%). Breathlessness and insomnia were more common in comparison to long-COVID group in our cohort.[34] Dyspnoea was also more common (34.4%) in the study of Moreno-Pérez et al. in 8-12 weeks observation in relation to 4-12 week observation of our cohort.[35]’

Point 5:use the terms people or individuals with MS instead of patients with MS/MS patients

Response 4: Thank you very much for your valuable comment. We changed the used term ‘patients with MS’ for ‘individuals with MS’ or ‘people with MS’ as suggested. In some general description parts the word ‘patients’ was left, we hope it is acceptable this way.
